# Resilience of constituent solitons
# in multisoliton scattering off barriers

V. Dunjko[1*], M. Olshanii[1]

**1** Department of Physics, University of Massachusetts Boston, Boston, MA 02125, USA

*dunjko.vanja@gmail.com

February 5, 2019

## Abstract

We introduce "superheated integrability," which produces characteristic staircase transmission plots for barrier collisions of breathers of the nonlinear Schrödinger equation. The effect makes tangible the inverse scattering transform, which treats the velocities and norms of the constituent solitons as the real and imaginary parts of the eigenvalues of the Lax operator. If all the norms are much greater than the velocities, an integrability-breaking potential may nonperturbatively change the velocities while having no measurable effect on the norms. This could be used to improve atomic interferometers.

## 1  Introduction

Completely integrable partial differential equations (PDEs) and solitons have played a central conceptual role in studies of nonlinear dynamics, especially of integrability-to-chaos transition and thermalization [1]. Moreover, these equations often model real systems. A very well-known example of an integrable PDE that supports solitons is the 1D focusing nonlinear Schrödinger equation (NLSE),

$$i\partial_t\psi = -\frac{1}{2}\partial_z^2\psi - |\psi|^2\psi + V(z)\psi\,, \tag{1}$$

integrable when $V = $ const. This equation applies to light propagating in nonlinear planar waveguides and optical fibers [2]; small-amplitude gravity waves on the surface of deep inviscid water [2]; the Langmuir waves in hot plasmas [2]; storage and transfer of vibrational energy in $\alpha$-helix proteins via Davydov's solitons [3]; and, of special relevance to us, ultracold quantum Bose gases, in the mean-field regime, in highly elongated traps that make them effectively one-dimensional [4]. In particular, since their first experimental realizations [5–7], bright matter-wave solitons have also been studied in the context of matter-wave interferometry. Their use may improve sensitivity by several orders of magnitude [8, 9], especially in precise force sensing [9, 10] and in the measurement of small magnetic field gradients [9, 11]. Much-studied are the soliton collisions with potential barriers, resulting in splitting [12–19] and subsequent recombination of solitons [20, 21].

   In this work we present, in the context of the focusing 1D NLSE, an effect we call 'superheated integrability', which highlights the inverse scattering transform (IST) structure of the problem: a multisoliton's constituent solitons can survive a temporary *non-perturbatively strong* integrability breaking. Our main setting is multisoliton collisions with a barrier, but it may also be observed in other circumstances.

## 2  The setting of the problem

Consider first the scattering of an NLSE (ground-state) 1-soliton, $\psi = \frac{1}{2}\exp(it/8)\operatorname{sech}(z/2)$, in a regime which *energetically* protects it from splitting or radiating its norm away [22]. This protection exists even for solitary waves of *nonintegrable* PDEs, but here is the argument for the case of the 1D NLSE in the context of matter waves (see Appendix A). Let $v_{\mathrm{CM}}$ be the velocity of the soliton

center of mass; $N_\mathrm{a}$, the total number of atoms in the soliton, each of mass $m$; and $E_\mathrm{rest}/N_\mathrm{a}$, the soliton energy per particle in the center-of-mass rest frame. Note that $E_\mathrm{rest}$ is negative and proportional to $(N_\mathrm{a})^3$ (see Appendix B). The regime in which the soliton is energetically protected from particle loss is $|E_\mathrm{rest}|/N_\mathrm{a} \gg E_{\mathrm{K},1} = \frac{1}{2}mv_\mathrm{CM}^2$. The reason, as we show in Appendix B, is that the soliton is energetically allowed to lose $\Delta N$ particles during the barrier collision, but $\Delta N/N_\mathrm{a} \propto 1/N_\mathrm{a}^2$ if we increase $N_\mathrm{a}$ while keeping $v$ constant, and the number of particles $N_\mathrm{a}$ is assumed to be large, about $3 \times 10^4$ [21].

Now suppose that instead of a simple soliton, we scatter a *multisoliton* [1, 23] (also called a *bound state* [1, 23–25]). This is a 'nonlinear superposition' of two or more solitons propagating in such a way that their centers of mass coincide. Although the field is not a linear superposition of simple solitons (see e.g. Eq. (9) in Appendix G), the identity of the constituent solitons ('consolitons') is clear (see below and Appendix E). This is also a *breather* [23, 26] because the object oscillates in time. (Note that this is a bit different than in the case of the sine-Gordon equation, where breathers and 2-solitons are distinct objects [1].)

Let us consider the 1:3 breather, whose consolitons have norms $1/4$ and $3/4$. This 2-soliton is of special interest to us because it can be created from the basic soliton by a quench in the coupling constant (a property it shares with all 'odd-norm-ratio breathers' (ONRBs), as we will explain below). Its 'breathing' is shown in Fig. 3 in Appendix G. Let us assume that the regime is such that if either of the two consolitons is scattered individually, it would be energetically protected from particle loss. However, since these two solitons are in fact overlapping, we should note that it is *energetically favorable* for the larger one to absorb particles from the smaller one. Since the interactions *within* and *between* the consolitons are of the same order, such a transfer of particles might occur as soon as integrability is significantly broken, e.g. when the multisoliton is significantly straddling a large barrier. So a *naive prediction* would be that when the 2-soliton is scattered off a barrier, the outgoing field should still consist principally of two consolitons, but whose norms have changed: the larger consoliton should have become even larger, and the smaller one, smaller.

## 3 The main result

Our main result is that the consolitons can in fact be *preserved*—even though the process is *strongly nonperturbative* in the standard sense of the word: the field at the beginning is *very* different from that at the end.

In our numerics, we initialize a 2-soliton with consolitons of norms 1/4 and 3/4 (Appendix G). We launch it (using a Galilean boost, Appendix D) to collide with a barrier $V(z) = V_0 \exp(-2(z/w)^2)$, whose width puts it in between a substantially quantum (delta-function-like) and semiclassical regimes. (A comparably sized rectangular barrier gives similar results). Figure 1 shows that, with our chosen impact speed, the 2-soliton always splits into two well-separated 1-solitons. We verified they have norms 1/4 and 3/4 to several significant figures (see also Fig. 2). Thus, in our regime, for all intents and purposes, the transmission of each individual consoliton is all or nothing. For a high barrier, both splinters are reflected. We did the following series of numerical experiments: in each experiment, we scatter our 2-soliton off a barrier. The impact velocity is the same for all experiments, but the barrier heights are different in each experiment. The first experiment has a very high barrier, and the result of the scattering is two splinter solitons, of norms 1/4 and 3/4, being reflected off the barrier. Now, in each successive experiment, we lowered the barrier height by a little bit while keeping all other parameters the same. For a while, this lowering of the barrier height had no qualitative effect: both splinters

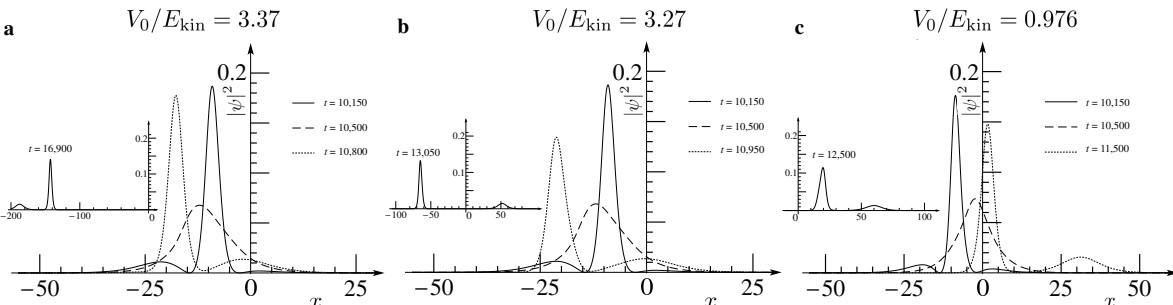

Figure 1: **Three kinds of outcome following a collision of a 1:3 ONRB with a Gaussian barrier.** All numbers are in natural units (see Appendix C). The barrier is at $x = 0$, has peak height $V_0$, and $1/e^2$ half-width $w = 9.5$. The breather has impact speed of 0.025, and chemical potential which is much higher than its (center-of-mass) kinetic energy per particle. The main plots show the details of the collision process, where the various density profiles correspond to different times $t$; the insets are the final outcomes. **a,** total reflection; **b,** the constituent soliton ('consoliton') of norm 1/4 is transmitted, while the soliton of norm 3/4 is reflected; **c,** both consolitons are transmitted. The underlying experimental values are as in the current lithium experiments, except for the impact velocity and barrier width, which are ten times lower and larger, respectively. This makes the ratio of the chemical potential to kinetic energy per particle large enough that the changes in consoliton norms are unmeasurable. In the final outcomes, the consolitons are well-separated even when they are on the same side of the barrier because they generally spend unequal times on top of the barrier and leave the barrier with unequal speeds.

kept getting reflected. However, there was a critical height at which the smaller splinter started to be transmitted, while the larger one still kept getting reflected. Further lowering of the barrier height then again, for a while, had no qualitative effect: the smaller splinter kept getting transmitted, and the larger kept getting reflected. Finally, as the barrier was lowered even further, there was another critical height such that the larger splinter started to be transmitted as well, and from there on both splinters were getting transmitted. We found that the exact values of the critical heights depend on the details such as the type of barrier (e.g. Gaussian vs. square), the barrier width (see Fig. 2 below and Fig. 4 in Appendix I), and even the phase during the breathing cycle at which the collision occurs (see Fig. 6 in Appendix I). What is universal, however, is that (i) there *are* critical heights and (ii) the splitters are always of norms 1/4 and 3/4 (to several significant figures), with the smaller consoliton having the larger critical height. This results in the staircase-like transmission plots of Fig. 2, where the stair widths are *not* universal but the stair heights (1, 1/4, and 0) *are*. Decompositions of breathers into consolitons have been studied previously, but only for weak integrability-breaking perturbations [23, 26].

It may seem unusual that, in our series of experiments, we are keeping the impact velocity constant while varying the height of the scattering potential, rather than the other way around. But this is natural in our case because the impact velocity enters the principal governing parameter of the problem: the ratio between the breather's energy-per-particle in its center-of-mass frame ($E_{\rm rest}/N_{\rm a}$) and the kinetic energy, per particle, of the center of mass $E_{\rm K,1}$. The larger this parameter, the better are the consolitons energetically protected against shedding particles during the barrier collision.

Our 1:3 two-soliton is the first in a sequence of odd-norm-ratio breathers (ONRBs) [25], whose $n$th element is an $n$-soliton with consolitons of norm ratios $1 : 3 : \cdots : (2n - 1)$. Like all multisolitons whose constituent breathers have unequal norms (see Appendix F), ONRBs periodically beat in space due to interference and the fact that the angular velocities of consoliton phases depend on their norm. Experimentally important is that once per breathing period, an ONRB's waveform is a sech, the same as the ground-state soliton for an NLSE *with a different coupling constant* ($g_{\rm 1D}$ in Eq. (4) in Appendix A), one $n^2$ times smaller in magnitude.

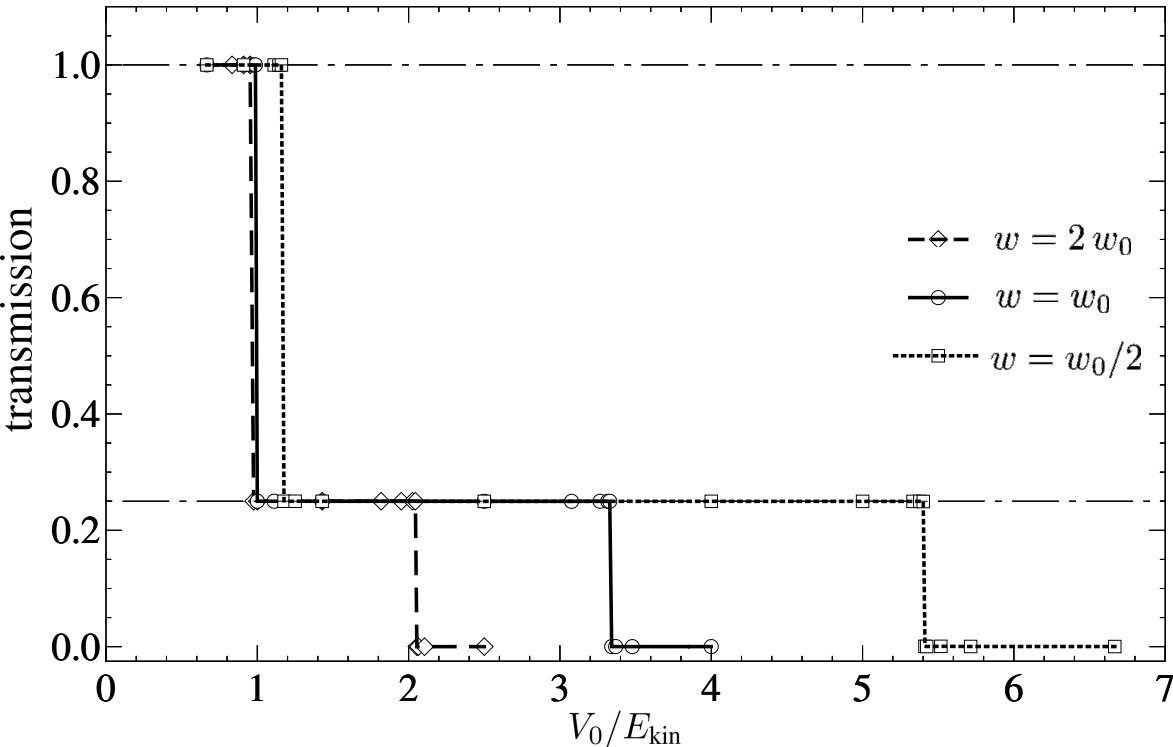

Figure 2: **Transmission plot for the scattering of a 1:3 ONRB off a Gaussian potential.** The dash-dotted horizontal lines are at transmissions values of 1/4 and 1, corresponding, respectively, to only the smaller soliton being transmitted, and to both solitons being transmitted. The three curves correspond to three different values of barrier width $w$; in all cases, as a function of $V_0/E_{\text{kin}}$, the transmission proceeds in the same three steps **a**, **b**, and **c** presented in Fig. 1, with a broad plateau during which the transmission is 1/4. All other parameters are as in Fig. 1. The effect persists for at least a factor of 15 of variation in impact velocity and a factor of 200 of variation in barrier width (Fig. 4 in Appendix I). Figure 8 in Appendix I shows how this plot would look if the constituent solitons were completely decoupled.

A breather (with any number of consolitons) is sometimes referred to as a bound state of its consolitons [24]. However, the dissociation energy is zero; this is because there is a fine-tuned cancellation of the internal potential and kinetic energies, both of which are individually substantial (so the coupling is strong) [23, 26]. Let us explain that statement on the example of the 2-soliton. Let $v_1$ and $v_2$ be the velocitites, respectively, of the two consolitons (see Appendix E). If $|v_1 - v_2| = \epsilon > 0$, then for $t \gg 1/\epsilon$, the solitons will be well-separated and ballistically receding from each other with a speed that tends to $\epsilon$ as $t \to \infty$.

A few points: (i) The effect persists for at least a factor of 15 of variation in impact velocity and a factor of 200 of variation in barrier width (Fig. 4 in Appendix I). (ii) During the collision, the field over the barrier is never small, so the nonlinear interactions are never negligible (Fig. 1). (iii) The stable plateaus in the transmission plots disappear if the underlying equation is not integrable (Fig. 5 in Appendix I). (iv) The phase of the breathing cycle at which the collision occurs affects where the plateau at 1/4 begins and ends, but not its existence (Fig. 6 in Appendix I). (v) The effect generalizes to $n$-solitons (Fig. 7 in Appendix I).

# 4 An explanation for the effect

Here is why the effect happens. Given the NLSE of Eq. (1), the inverse scattering transform (IST) introduces, for each time $t$, an auxiliary linear problem for functions $u(z, t)$ and $v(z, t)$; the NLSE field $\psi(z, t)$ plays the role of a potential:

$$
\begin{pmatrix} -i\partial_z & \psi^* \\ -\psi & i\partial_z \end{pmatrix} \begin{pmatrix} u \\ v \end{pmatrix} = \lambda \begin{pmatrix} u \\ v \end{pmatrix} .
\tag{2}
$$

The square matrix is called the Lax operator. For a given $\psi$, the problem produces *scattering data*: 1. the reflection coefficient, strictly zero if $\psi$ consists purely of solitons; 2. the normalization coefficients (the prefactors of the long-distance asymptotics of bound states); and 3. the discrete eigenvalues $\lambda$. Conversely, given scattering data, one can recover $\psi$ by solving the inverse scattering problem. Similar linear problems may be introduced for both integrable and nonintegrable PDEs; but only the former have that [27]: 1. the scattering data at $t = 0$ may be *easily* propagated to any $t$; and, 2. the $\lambda$s are *time independent*. Thus, a method of solution: get the initial scattering data from $\psi$ at $t = 0$; propagate the scattering data to some desired $t$; solve the inverse scattering problem for this propagated data (a linear integral equation, analytically tractable in many physically important cases), obtaining $\psi$ at $t$. The IST is reminiscent of how the Fourier transform is used to solve *linear* PDEs, where the IST scattering data play the role analogous to the Fourier coefficients.

Each distinct $\lambda_j$ describes a consoliton: in the natural units (Appendix C), $\lambda_j = A_j/2 + iv_j$, where $v_j$ is the consoliton velocity and $A_j$ its norm (see Eq. (8) in Appendix E). Breathers have consolitons with the same velocities and initial positions.

With an integrability-breaking potential $V(z) = \epsilon\, v_{\text{ext.}}(z)$, where $v_{\text{ext.}}(z) \sim 1$ and $\epsilon$ positive but not (yet) assumed small, the $\lambda_j$'s develop time dependence. This is described by certain well-known relations of so-called IST perturbation theory [27], which normally involve bound-state asymptotics. However, we prefer the following exact relationship which instead involves just the normalization integral $\nu = 2 \int_{-\infty}^{\infty} dz\, u\, v$ (for simplicity of notation, we are suppressing the subscript $j$; in principle it should be $\lambda_j$, $u_j$, $v_j$, and $\nu_j$):

$$
\dot\lambda = (i/\nu)\,\epsilon \int_{-\infty}^{\infty} dz\, v_{\text{ext.}} \left( u^2\, \psi + v^2\, \psi^* \right) ,
\tag{3}
$$

where $\psi$ is the *exact* solution of the *perturbed* NLSE, and $\dot\lambda = d\lambda/dt$. The total change is $\Delta\lambda = \int_{-\infty}^{\infty} \dot\lambda \, dt$. Even though we are unaware of it appearing in print previously, Eq. (3) is probably already known to specialists. However, we invented a new, simple derivation of it based on the Hellmann-Feynman theorem (see Appendix H).

Superheated integrability obtains when $|\mathrm{Re}\,\lambda| \lesssim |\Delta\lambda| \ll |\mathrm{Im}\,\lambda|$ for all $\lambda$s corresponding to consolitons of a breather. If this holds, then the soliton identity is preserved—the imaginary parts of the $\lambda$s (soliton norms) remain unaffected by the collision—even while their real parts (soliton velocities) change substantially. To see how this can happen, let us compute the scalings with $\epsilon$ in a regime where the impact velocity is small and the breather is wider than the barrier. In our regime, the norm of the breather (and so of its consolitons) is $\sim 1$. Its impact kinetic energy is of the same order as the potential, so the impact velocity $\sim \sqrt{\epsilon}$. We assumed that the impact velocity is small, so $\epsilon \ll 1$. The barrier is important only while the breather is on top of it. The following seems physically reasonable even though we could not rigorously justify it: *even while the consolitons are not well separated*, as long as the norms of the consolitons are not much different from their initial values, the centers of mass of the consolitons will, at least qualitatively, move as if the other consoliton(s) wasn't (weren't) there (Fig. 8 in Appendix I illustrates this principle). Thus we have the estimate that for each consoliton, $\Delta\lambda \sim \dot\lambda \times$ typical collision time, and $\dot\lambda \sim \epsilon$. To estimate the typical consoliton collision time, note that since the barrier is narrower than the breather, the spatial extent along which the barrier has a significant effect is the same as the width of the breather, i.e. $\sim 1$. Thus the time that consolitons spend atop the barrier scales as (breather width)/speed $\sim 1/\sqrt{\epsilon}$. Therefore, $\Delta\lambda \sim \sqrt{\epsilon}$.

Thus, $\lambda = A/2 + iv$, with velocity $v \sim \sqrt{\epsilon}$ and the norm $A \sim 1$. Therefore, $\Delta\lambda$ is comparable in magnitude to the velocity $v$, but is much smaller than the norm $A$. Thus the norm remains unaffected even as the velocity changes substantially—precisely our effect.

The above estimate of the collision time fails when one of the consolitons has the velocity close enough to critical (i.e. between it being completely reflected and completely transmitted) to be spending a long time atop the barrier. But the consolitons all have different norms and thus different critical velocities. So the time estimate can only fail for *one* of the consolitons. The others will therefore leave the barrier in the time $\sim 1/\sqrt{\epsilon}$, while the remaining one lingers. However, since it is now a single well-separated soliton, it is energetically protected from changing its norm (Appendix B), and so we again have our effect.

In summary, as the breather begins to climb the barrier, the changes in the norms of the consolitons (imaginary parts of their $\lambda$s) slowly begin to accumulate according to Eq. (3). But the solitons will leave the barrier before their norms can change significantly—with a possible exception of exactly one of the solitons having just the right size-velocity combination to linger atop the barrier. But this soliton is then energetically protected from fragmenting.

We should note that the effect persists even in regimes where some of the assumptions we made above are not valid, e.g. even when the barrier is much wider than the breather (see Fig. 4 in Appendix I). Accounting for the full range of parameters across which our effect exists is an open problem that will require further study of the IST perturbation theory [27].

In the usual sense of the word, the process is heavily nonperturbative, since it results in a dramatic change in the shape of the field $\psi$. Save for the norms, all of the consoliton parameters change dramatically. If we metaphorically picture the perturbation as a kind of 'heating up' of the previously conserved ('frozen') quantities, the norms appear 'superheated': though they 'should' start flowing, they do not. The reason is that the IST, counterintuitively, treats the soliton velocity and norm as a *single entity*, as the real and imaginary parts of a coordinate $\lambda$ in the 'IST space' (a nonlinear counterpart to the momentum space of the Fourier transform). The 'superheating' of the norm thus highlights the underlying IST structure.

# 5 A proposal for an experimental realization

Experimentally, the 1:3 breather can be excited by first creating the (ground-state) soliton, and then quenching the coupling constant up fourfold in magnitude. Since the soliton is usually first created very close to the stability threshold [28] and the breather may be more unstable than a single soliton (although perhaps it could be metastable more robustly), the magnitude of the coupling constant should first be adiabatically reduced eightfold, and only then quenched up fourfold [29]. In the $^7$Li experiments [21], the frequency of the transverse harmonic confinement is $\omega_r = 2\pi \times 254\,\text{Hz}$, while the residual longitudinal confinement is $\omega_z = 2\pi \times 31\,\text{Hz}$. The scattering length of the atomic interactions ($a$ in Eq. (5) in Appendix A) is $-1.0 \times a_0$ (where $a_0$ is the Bohr radius), but it can be tuned down in magnitude at least tenfold by using the magnetic Feshbach resonance [21]. The soliton has $N_a = 3 \times 10^4$ atoms, which is indeed at the collapse threshold. The barrier is a $900\,\text{GHz}$ blue-detuned Gaussian laser beam whose half-width at the $1/e^2$ intensity point is $4.2\,\mu\text{m}$, while the barrier height is anywhere from $0.2\,\text{kHz}$ to $3\,\text{kHz}$ (times $h$, the Planck's constant). The typical soliton impact speed in current experiments is $514\,\mu\text{m/s}$. In our numerical experiments, we took $a = -0.41 a_0$; in order to produce very sharp transitions in the transmission plots, we lowered the impact velocity tenfold from the currently typical experimental value. For the smaller soliton, this resulted in $(E_{\text{rest}}/N_a)/E_{\text{K},1} \approx 8$. Stable plateaus can be obtained even with larger velocities, though the transitions will no longer be as sharp (e.g. the dashed line in Fig. 5 in Appendix I). Also, to limit the computational grid, we increased the barrier width tenfold from the experimental value (results of Fig. 2 suggest this will not matter qualitatively).

# 6 Conclusion

The essential ingredients of superheated integrability are (i) a separation of scales in the magnitudes of the real and the imaginary parts of the eigenvalues of the Lax operator; and (ii) an integrability-breaking perturbation whose effect is, for each eigenvalue $\lambda$, greater than or comparable to the smaller of the pair ($|\text{Re}\lambda|, |\text{Im}\lambda|$), but much smaller than the greater of that pair. The nontrivial aspect of predicting or explaining an occurrence of superheated integrability is estimating the magnitude of the effect of the perturbation. But it should be clear that there must exist other types of superheated integrability, in other settings, even in other integrable systems. In fact, we can already give another example of superheated integrability in the NLSE, in which integrability is broken not by a collision with a barrier, but by phase imprinting; see Fig. 9 in Appendix I.

The existence of a transmission plateau as in Fig. 2 could be used to stabilize bright solitonic atom interferometers against the fluctuations of the profile of the beam splitter potential, and against the parasitic mean-field effects [8,9]. In the latter case, the problem is that due to interatomic interactions, the amount of phase that a matter wave accumulates as it travels depends also on its size. But the current interferometric schemes rely on splitting a single soliton off a barrier, and so a lack of control over what portion of atoms goes to each interferometer arm results in an accumulation of different amounts of phases (which one does not know how to account for) in each arm during free flight [8]. For flight times that will be necessary to produce competitive accelerometers, the lack of control present in current schemes will be unacceptably large. An interferometric scheme that takes advantage of the superheated integrability of soliton norms, however, may overcome this problem.

## Acknowledgements

We thank B. Sundaram, B. A. Malomed, R. Hulet, and P. Dyke for useful discussions; we additionally thank R. Hulet and P. Dyke for providing us with detailed parameters of their experiments.

**Funding information**  This work was supported by grants from the Office of Naval Research (N00014-12-1-0400) and the National Science Foundation (PHY-1402249).

## A  The 1D nonlinear Schrödinger equation in the context of ultracold Bose gases

In the context of ultracold Bose gases, the 1D nonlinear Schrödinger equation, also called the 1D Gross-Pitaevskii equation (GPE) in that setting, is

$$i\hbar\partial_t\Psi = -\frac{\hbar^2}{2m}\partial_z^2\Psi + g_{1\mathrm{D}}N_\mathrm{a}|\Psi|^2\psi + \tilde{V}(z)\Psi\,, \tag{4}$$

where $\int_{-\infty}^{\infty}|\Psi(z,t)|^2\,dz = 1$. It describes the dynamics of the order parameter $\Psi(z,t)$ of the Bose-Einstein (quasi)condensate of $N_\mathrm{a}$ atoms of mass $m$ confined to a cigar-shaped waveguide, in which the transverse confinement (a harmonic potential, $\frac{1}{2}m\omega_r^2 r^2$) is so tight that the energy spacing $\hbar\omega_r$ between the ground and the first excited transverse state is much greater than the typical energy of the confined particles, making the system effectively 1D. The atoms interact with an effective pairwise interaction $g_{1\mathrm{D}}\,\delta(z_j - z_k)$, where $z_j$ and $z_k$ are the positions of the atoms. Here the 1D coupling constant is given by [30–32]

$$g_{1\mathrm{D}} = 2\hbar\omega_r a\,, \tag{5}$$

where $a$ is the scattering length of the atom-atom interactions (which in our case will be negative). More precisely, it is [30, 31]

$$g_{1\mathrm{D}} = \frac{2\hbar^2 a}{\mu a_r^2}\frac{1}{1 - Ca/a_r}\,,$$

where $\mu = m/2$ is the reduced mass, $a_r = \sqrt{\hbar/(\mu\omega_r)}$ is the width of the ground state of the transverse confinement in relative coordinates, and $C = -\zeta(1/2) \approx 1.46035\ldots$, where $\zeta$ is the Riemann zeta function. However, for us, the magnitude of $a/a_r$ is on the order of $10^{-5}$ (see below), and so for all intents and purposes, $g_{1\mathrm{D}} = 2\hbar^2 a/\left(\mu a_r^2\right) = 2\hbar\omega_r a$, which is Eq. (5). This means that the coupling constant is, with negligible corrections, proportional to the scattering length. We need $g_{1\mathrm{D}} < 0$ for the system to support bright solitons, so $a < 0$. The number of atoms $N_\mathrm{a}$ is assumed large enough that we expect that quantum corrections to be negligible [4, 8] (this is also why we ignore the fact that it really should enter as $N_\mathrm{a} - 1$). The potential $\tilde{V}(x)$ is our barrier, in experiments simply a blue-detuned Gaussian laser beam: $\tilde{V}(z) = \tilde{V}_0\exp(-2(z/\tilde{w})^2)$, where $\tilde{w}$ is the half-width of the laser beam at the $1/e^2$ intensity point. Note that the central intensity of the beam, $\tilde{V}_0$, is usually reported as a frequency (in hertz). It is numerically equal to $\tilde{V}_0/h$, where $h$ is Planck's constant. In principle, one should also add to $\tilde{V}(z)$ a residual longitudinal harmonic trapping, characterized by the frequency $\omega_z$.

The 1D mean-field theory will describe well the longitudinal dynamics of the elongated 3D condensate if $|a|N_\mathrm{a}|\Psi|^2 \ll 1$ [32]. On the other hand, for $a < 0$, if the product $|a|N_\mathrm{a}$ is too large, the system is unstable against collapse. More precisely [28], for highly elongated condensates and $a < 0$, a stable ground state exists only for $|a|N_\mathrm{a}/a_\perp < 0.676$, where $a_\perp = \sqrt{\hbar/(m\omega_r)}$.

The relevant values from $^7$Li experiments at Rice [21] are: $\omega_r = 2\pi \times 254\,\text{Hz}$, $\omega_z = 2\pi \times 31\,\text{Hz}$, $a = -1.0 \times a_0$ (where $a_0$ is the Bohr radius), $N_a = 3 \times 10^4$, $\tilde{w} = 4.2\,\mu\text{m}$, and $\tilde{V}_0/h$ is anywhere from $0.2\,\text{kHz}$ to $3\,\text{kHz}$. We see that $|a|N_a/a_\perp = 0.667$, so the system is very close to the stability threshold. This is why the first step after the ground state soliton is created must be an adiabatic reduction of the magnitude of $a$, most likely by a factor of eight [29].

## B   Energetically allowed fractional loss of particles for a single soliton

Consider the stationary, single-soliton solution (normalized to 1) of Eq. (1) with $V = 0$ in the main text, $\Psi_1(z) = \sqrt{c/2}\,\text{sech}(cz)$, where $c = |g_{1D}|N_a m/(2\hbar^2)$.

The energy-per-particle of the soliton is given by

$$E/N_a = \int_{-\infty}^{\infty} dz \left( \frac{\hbar^2}{2m}|\partial_z \Psi_1(z)|^2 + \frac{g_{1D}N_a}{2}|\Psi_1(z)|^4 \right)$$

$$= -dN_a^2, \text{ with } d = \frac{mg_{1D}^2}{24\hbar^2},$$

where $g_{1D} < 0$. (A related quantity, the chemical potential, is given as $\mu = \partial E/\partial N_a = -3dN_a^2$.) A fracturing of a single soliton into two smaller ones, $N_a \to N_1 + N_2$, raises the energy-per-particle by $2dN_1 N_2$. For multiple fragments, the energies are even less negative, so the increase in the energy-per-particle is even greater; same if some of the particles are radiated away, as their energy will actually be positive. So the assumption of a fracture into exactly two smaller solitons will give the maximal possible particle loss. If the fracturing is to occur due to the soliton (with center-of-mass speed $v$) colliding with a barrier, this energy can come only from the kinetic energy per particle of the center of mass, $E_{K,1} = \frac{1}{2}mv^2$. Let us write $N_1 = N_a - \Delta N$ and $N_2 = \Delta N$, where, without loss of generality, we impose the condition that $0 \leqslant \Delta N < N_a/2$. We therefore have $0 \leqslant 2d(N_a - \Delta N)\Delta N \leqslant E_{K,1}$; dividing through by $2dN_a^2$, we get

$$0 \leqslant \frac{\Delta N}{N_a} - \left(\frac{\Delta N}{N_a}\right)^2 \leqslant \frac{E_{K,1}}{2dN_a^2}. \tag{6}$$

Now we invoke the assumption that $E_{K,1} \ll |E/N_a| = dN_a^2$. Thus the right-hand side of Eq. (6) is much less than one. Next we show that $\Delta N/N_a$ is then itself much less than one. Let $x = \Delta N/N_a$, and let $\epsilon = x - x^2$. We have just established that $0 < \epsilon \ll 1$. Now we express $x$ in terms of $\epsilon$. First we note that the two solutions of $x - x^2 = \epsilon$ are $x_{1,2} = \frac{1}{2} \pm \sqrt{\frac{1}{2} - \epsilon}$. One of them is close to zero, the other close to one. But because we imposed that $\Delta N < N_a/2$, it follows that $x = \Delta N/N_a$ must be much less than one, as claimed. Because of that, we may neglect the quadratic term in Eq. (6), obtaining

$$0 \leqslant \frac{\Delta N}{N_a} \leqslant \frac{E_{K,1}}{2dN_a^2}.$$

See also Ref. [22] (the paragraph before Sec. 3).

## C   Conversion to natural units

By choosing to work in "natural units," Eq. (4) may be written as,

$$i\partial_t\psi = -\frac{1}{2}\partial_z^2\psi \pm |\psi|^2\psi + V(z)\psi\,,$$

where the sign of the $|\psi|^2\psi$ term is the sign of $g_{1D}$. In the present work, this is negative (i.e. attractive, or 'focusing'), so that it can support bright solitons, and in that case we get Eq. (1) from the main text. In general, we can allow the field to be normalized to any $\mathcal{N} = \int_{-\infty}^{\infty} |\psi(z,\,t)|^2\,dz$. In the natural system of units, we should have $\hbar = m = 1$, which fixes the unit of length to $u_L = \mathcal{N}\hbar^2/(mg_{1D}N_a)$ and the unit of time to $u_T = \mathcal{N}^2\hbar^3/(mg_{1D}^2N_a^2)$ (the unit of mass is of course $u_M = m$). The field is transformed as $\Psi(z,\,t) = \psi(z/u_L,\,t/u_T)/\sqrt{\mathcal{N}u_L}$. The derived units are found through the usual dimensional formulas, e.g. the unit of energy is $u_E = u_M u_L^2/u_T^2 = \frac{1}{\mathcal{N}^2}\frac{m}{\hbar^2}g_{1D}^2N_a^2$.

## D   Galilean transformation and norm rescaling of the solutions of non-linear Schrödinger equation

We are considering the integrable focusing 1D NLSE,

$$i\partial_t\psi(z,\,t) = -\frac{1}{2}\partial_z^2\psi(z,\,t) - |\psi(z,\,t)|^2\psi(z,\,t)\,. \tag{7}$$

If $\psi(z,\,t)$, of norm $\mathcal{N} = \int_{-\infty}^{\infty} |\psi(z,\,t)|^2\,dz$, is a solution of this equation, then so are the Galilean-transformed solution $\exp[iv(z-z_0) - i\frac{1}{2}v^2t]\,\psi(z-vt-z_0,\,t)$ and the norm-rescaled solution $\xi\psi(\xi z,\,\xi^2 t)$ of norm $\xi\mathcal{N}$.

## E   Solutions of the nonlinear Schrödinger equation that consist purely of solitons

We will be discussing solutions of Eq. (7) consisting purely of $N$ solitons (i.e. with no radiation).

The stationary 1-soliton (i.e. the ground-state soliton), of unit norm, is $\frac{1}{2}e^{it/8}\,\mathrm{sech}(z/2)$. The transformations of Appendix D may be used to produce 1-solitons of arbitrary norm, initial position, and velocity.

An $N$-soliton solution, for $N > 1$, is never a linear superposition of 1-solitons, because Eq. (7) is nonlinear. However, it may approach such a linear combination asymptotically, in the limit as $t \to \pm\infty$; we then say that 'all the constituent solitons are well-separated at infinity'. Generic solitonic solutions are of this form; we will call them 'separating' solutions.

Nevertheless, there are also cases which may be described as consisting of two or more 1-solitons that are never well-separated, even at infinite times; we will call them 'non-separating' solutions. Below we will explain why it nevertheless makes sense to talk of 'constituent 1-solitons' in this case, even though such solutions are not in any sense linear superpositions of 1-solitons, in any limit. Indeed, as will become evident below, in this case it makes sense to talk of 'nonlinear superpositions of 1-solitons'. These never-separated groups of solitons are nonlinear superpositions of 1-solitons that all have the same velocity. If in addition the constituent 1-solitons also have the same initial positions, so

that they propagate one on top of the other, then they are called either multisolitons, or bound states of solitons, or breathers (the latter because the field density profile undergoes periodic oscillations; see e.g. Fig. 3 in Appendix G). Special treatment is needed when the constituent 1-solitons have the same velocity as well as the same *norm*; this *degenerate* case is qualitatively different from the rest, and is briefly discussed in Appendix F. There we also explain why this class of solutions is of no interest to us.

The above distinctions notwithstanding, all $N$-soliton solutions may be parametrized by $4N$ real parameters, four per constituent soliton [33]. For $j = 1, \ldots, N$, these are: $A_j > 0$ (the norm of the $j$th constituent soliton), $v_j$ (its center-of-mass velocity), $z_{j,0}$ (what its $t = 0$ position would be if the other constituent solitons were not present), and $\phi_{j,0}$ (what its initial phase similarly would have been). The norm of $\psi$ is $\sum_{j=1}^{N} A_j$. If the velocities $v_j$ are all distinct, then at $t \to \pm\infty$, the constituent solitons of $\psi$ all become well-separated from each other, and $\psi$ becomes a sum of $N$ 1-solitons, of norms $A_j$, traveling at velocities $v_j$. Because of this parametrization, non-separating solutions are clearly limiting cases of separating ones, as $v_j \to v_k$ for some $j$ and $k$. This is what justifies, for the non-separating solutions, the talk of 'constituent solitons' and of 'nonlinear superposition'.

Given these $4N$ parameters, any non-degenerate $N$-soliton solution may be constructed as follows [33]: we have that

$$\psi(z, t) = \sum_{j=1}^{N} u_j(z, t),$$

where the $u_j$'s are the solutions of the following linear system of $N$ equations with $N$ unkowns:

$$\sum_{k=1}^{N} \frac{1/\gamma_j + \gamma_k^*}{\lambda_j + \lambda_k^*} u_k = 1, \quad j = 1, \ldots, N,$$

where the stars ('*') denote complex conjugation,

$$\lambda_j = A_j/2 + iv_j \tag{8}$$

and

$$\gamma_j = \exp\left[\lambda_j(z - z_{j,0}) + i\lambda_j^2 t/2 + i\phi_{j,0}\right].$$

The $\lambda_j$'s are precisely the eigenvalues of the Lax operator; see Eqs. (12)-(16) in Appendix H.

If a soliton is well-separated from all the others (note that as $t \to \pm\infty$, if the velocities $v_j$ are all distinct, then all the solitons will become more and more spatially separated from each other), its form will converge to the 1-soliton one,

$$\frac{A_j}{2} \operatorname{sech}\left[\frac{A_j}{2}(z - z_j) + q_j\right] \exp[i(\phi_j + \Psi_j)],$$

where

$$z_j = z_{j,0} + v_j t,$$

$$\phi_j = v_j(z - z_j) + \frac{1}{2}\left(A_j^2/4 + v_j^2\right) t + \phi_{j,0}.$$

The real-valued quantities $q_j$ and $\Psi_j$, which are piecewise-constant functions of time, are given through

$$q_j + i\Psi_j = \sum_{\substack{k=1 \\ k \neq j}}^{N} \operatorname{sign}(z_k - z_j) \ln \frac{A_j + A_k + 2i(v_j - v_k)}{A_j - A_k + 2i(v_j - v_k)},$$

where sign $z$ is -1, 0, or +1 if $z < 0$, $z = 0$, or $z > 0$, respectively. Both $q_j$ and $\Psi_j$ are zero if the other solitons are not there (i.e. if $A_k = 0$ for all $k \neq j$). They account for the interaction with the other solitons, as will be explained shortly. The quantity $z_{j,0} - 2q_j/A_j$ is the 'extrapolated' $t = 0$ position of the soliton. If the $j$th soliton is isolated at several time segments, this extrapolated $t = 0$ position will be different for any two time segments between which the soliton underwent a collision. (Similar changes happen for the 'extrapolated' $t = 0$ phase, $\phi_{j,0} + \Psi_j$.)

In practice, if one wants to compute a solution $\psi$ so that some constituent solitons are, at $t = 0$, well-separated from the others and located at some desired positions with some desired phases, one relies on the fact that $q_j$ and $\Psi_j$ depend only on the ordering of the spatial positions of the solitons at the time of interest (here $t = 0$), and not, say, on the magnitudes of the relative distances. It follows that if a soliton is well-isolated solitons at $t = 0$ and one wants it to sit at $\bar{z}_{0,j}$ at $t = 0$, one may determine the appropriate value of $z_{j,0}$ as follows: temporarily set each $z_{j,0}$ to $\bar{z}_{0,j}$, and compute the $q_j$'s. The required actual $z_{j,0}$ is then $z_{j,0} = \bar{z}_{0,j} + 2q_j/A_j$, and one can now proceed to compute the $N$-soliton solution in full; a similar correction may be applied for the initial phases.

To see why an isolated soliton's interactions with the other solitons are accounted for by $q_j$ and $\Psi_j$, recall that when a soliton collides with another soliton, it (famously) emerges undisturbed except that it is further along its trajectory than it would have been had the collision not taken place. An intuitive picture of this process is as follows. Since the interactions are attractive, then, from the point of view of any individual soliton, the collision resembles the situation when the soliton drops into a potential well and then emerges from it. It turns out that this potential well is always such that the soliton velocity during the collision is higher than either before or after the collision, and so the soliton moves further along than it otherwise would have.

## F  The case of degenerate eigenvalues of the Lax operator

It is possible for two or more constituent solitons to have both the same norm and the same velocity, so that the corresponding Lax eigenvalues $\lambda_j$, Eq. (8), are degenerate. This case may be treated by finding the solution for the case when the eigenvalues are different, and then taking the limit as they become the same. If in addition these solitons have the same $z_{j,0}$'s, then, through this limiting procedure, one obtains solutions that are qualitatively different from those discussed thus far. For example, in the two-soliton case, as $t \to \pm\infty$, one finds [24] that the distance between the solitons increases proportionally to $\ln(A^2 t)$. Therefore the solitons separate on their own on the time scale of $1/A^2$, and so the collision experiment should last shorter than that. On the other hand, the breather needs to start sufficiently far from the barrier so that it begins in an approximately integrable regime, and it needs to be sufficiently slow so that the kinetic energy per particle is much less than the chemical potential. It turns out that these constraints are impossible to satisfy simultaneously, and thus the degenerate case is not of interest for us.

## G  The two-soliton

The 2-soliton odd-norm-ratio breather (ONRB) has the form

$$\psi_{\text{ONRB}[2]}(x, t) = \frac{e^{i\frac{t}{128}} \left( \cosh\frac{3x}{8} + 3\, e^{i\frac{t}{16}} \cosh\frac{x}{8} \right)}{6\cos\frac{t}{16} + 8\cosh\frac{x}{4} + 2\cosh\frac{x}{2}} ; \tag{9}$$

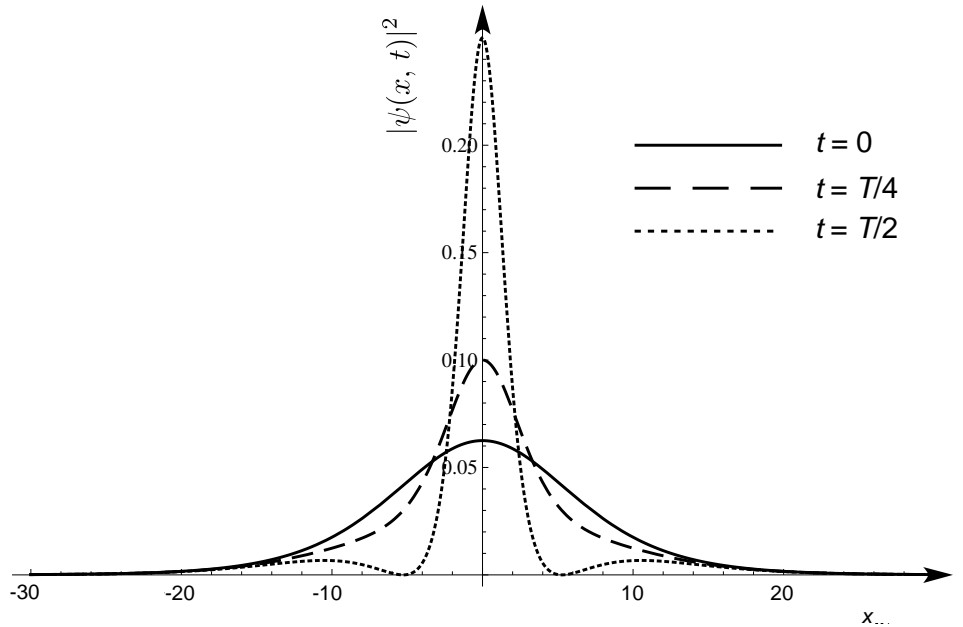

Figure 3: **The 1:3 odd-number ratio breather [see Eq. (9)], at three points of time.** Here $T = 32\pi$ (in natural units; see Appendix C) is the period of density oscillations. This is a particular two-soliton solution of the NLSE, where the constituent solitons have coinciding centers and zero velocities; the norms of the constituent solitons are 1/4 and 3/4.

it may be obtained from the general $N$-soliton solution discussed in Appendix E above if one sets $N = 2$, $A_1 = 3/4$, $A_2 = 1/4$, $\phi_{1,0} = 0$, $\phi_{2,0} = \pi$, and $v_1 = v_2 = z_{1,0} = z_{2,0} = 0$. The density profile is shown in Fig. 3 on p. 14.

In numerical scattering experiments, this 2-soliton is set in motion by using the Galilean boost described at the beginning of Appendix E.

## H  Derivation of the exact expression for $d\lambda/dt$, Eq. (5) in the main text

We are dealing with the 1D nonlinear Schrödinger equation, Eq. (4) in Appendix A above,

$$i\hbar \frac{\partial}{\partial t}\Psi(z, t) = -\frac{\hbar^2}{2m}\frac{\partial^2}{\partial z^2}\Psi(z, t) + g_{1D}N_a|\Psi(z, t)|^2\Psi(z, t) + \tilde{V}(z, t)\Psi(z, t), \quad (10)$$

where $g_{1D} < 0$, *in the presence of* the external integrability-breaking potential $\tilde{V}(z, t)$ (note that here we will allow this external potential to explicitly depend on time). In order to facilitate comparison with Ref. [27], which treats the same kind of problem, we will work in the units in which $\hbar = 1$, $m = 1/2$, and $|g_{1D}|N_a = 2$. Thus the NLSE becomes

$$i\frac{\partial}{\partial t}\psi(z, t) = \left[-\frac{\partial^2}{\partial z^2} - 2|\psi(z, t)|^2\right]\psi(z, t) + \epsilon v_{ext}(z,t)\psi(z, t), \quad (11)$$

where the external potential in (10) is factorized as

$$\tilde{V}(z, t) \equiv V_0 v_{ext}(z, t)$$
$$\max_z [v_{ext}(z, 0)] = 1 \,,$$

and the small parameter $\epsilon$ is

$$\epsilon \equiv \frac{2\hbar^2 V_0}{(g_{1D} N_a)^2 m} \,.$$

The first Lax operator reads

$$\hat{\mathcal{L}} = \begin{pmatrix} \hat{L} & \hat{M} \\ -\hat{M}^\dagger & -\hat{L} \end{pmatrix} \tag{12}$$

with

$$\hat{L} = -i\frac{\partial}{\partial z} \tag{13}$$

$$\hat{M} = \psi^*(z, t) \,. \tag{14}$$

For each instance of time $t$, one can set up an eigenstate-eigenvalue problem, which is the central object of interest of this derivation:

$$\hat{\mathcal{L}}|w\rangle = \lambda|w\rangle \,, \tag{15}$$

where

$$|w\rangle = \begin{pmatrix} u(z, t) \\ v(z, t) \end{pmatrix} \,. \tag{16}$$

To convert expressions in this text to the ones appearing in Ref. [27], one should use the following replacement table:

$$z \rightarrow x$$
$$\psi(z, t) \rightarrow u(x, t)$$
$$u(z, t) \rightarrow \psi^{(1)}(x, t)$$
$$v(z, t) \rightarrow \psi^{(2)}(x, t)$$
$$v_{ext}(z, t)\psi(x, t) \rightarrow i(P[u])(x, t) \,.$$

## H.1 Relevant functional analysis

From the fact that $\hat{L}$ is Hermitian, $\hat{L}^\dagger = \hat{L}$, it follows that the Lax operator (12) possesses the following property:

$$\hat{\mathcal{L}}^\dagger = \hat{\sigma}_3 \hat{\mathcal{L}} \hat{\sigma}_3 \,.$$

This property induces a particular Hermitian form $(\cdot, \cdot)$, a pseudo-inner product:

$$(|w_1\succ, |w_2\succ) = \prec w_1|w_2\succ \equiv \langle u_1|u_2\rangle - \langle v_1|v_2\rangle$$

$$= \int dz \left\{ u_1^*(z)u_2(z) - v_1^*(z)v_2(z) \right\}. \tag{17}$$

This Hermitian form lacks the property of being positive definite (i.e. lacks the property that, for all $|w\rangle$, $\langle w|w\rangle \geq 0$ and $\langle w|w\rangle = 0$ if and only if $|w\rangle = |0\rangle$). The rest of the inner product axioms, on the other hand, remain intact:

$$\prec w_2|w_1\succ = \prec w_2|w_1\succ^*$$

and

$$\prec w_1|aw_2 + bw_3\succ = a \prec w_1|w_2\succ + b \prec w_1|w_3\succ .$$

The Lax operator $\hat{\mathcal{L}}$ from Eq. (12) is symmetric with respect to this form:

$$\left(|w_1\succ, \hat{\mathcal{L}}|w_2\succ\right) = \left(\hat{\mathcal{L}}|w_1\succ, |w_2\succ\right) . \tag{18}$$

The property above justifies a standard notation $\left(|w_1\succ, \hat{\mathcal{L}}|w_2\succ\right) \equiv \prec w_1|\hat{\mathcal{L}}|w_2\succ$ that we are going to employ below.

The pseudo-Hermiticity property (18) implies the following properties of the eigenstates of $\hat{\mathcal{L}}$: Let $\hat{\mathcal{L}}|w_1\succ = \lambda_1|w_1\succ$, $\hat{\mathcal{L}}|w_2\succ = \lambda_2|w_2\succ$, and $\hat{\mathcal{L}}|w\succ = \lambda|w\succ$. Then

1. eigenvectors whose eigenvalues are not complex conjugates of each other are mutually orthogonal,

$$\lambda_1^* \neq \lambda_2 \;\; \Rightarrow \;\; \prec w_1|w_2\succ = 0 ;$$

2. non-zero norm eigenstates of $\hat{\mathcal{L}}$ correspond to real eigenvalues (a corollary of the above):

$$\lambda^* \neq \lambda \;\; \Rightarrow \;\; \prec w|w\succ = 0 .$$

Note that the eigenspectrum of $\hat{\mathcal{L}}$ is not necessarily complete. In cases when the Lax operator (12) represents a linear stability analysis equation of a nonlinear PDE, the missing states are associated with the continuous symmetries of the PDE that is broken by the solution in question [34]. (For a "flat" condensate, $\psi(z) = $ const, we found one missing state; there could be more. There seem to be none for a single soliton.)

The operator (12) also possesses properties specific to a particular form of the matrix elements, Eqs. (13) and (14):

$$\hat{L}^* = -\hat{L} .$$

This property implies that:

1. real eigenvalues $\lambda$ are doubly degenerate. The corresponding eigenstates,

$$|w\succ \doteq \begin{pmatrix} u(z) \\ v(z) \end{pmatrix}$$

$$|\tilde{w}\succ \doteq \begin{pmatrix} \tilde{u}(z) \\ \tilde{v}(z) \end{pmatrix} ,$$

are related by

$$\tilde{u}(z) = -v^*(z)$$
$$\tilde{v}(z) = +u^*(z) .$$

Here,

$$\hat{\mathcal{L}}|w\succ \doteq \lambda|w\succ$$
$$\hat{\mathcal{L}}|\tilde{w}\succ \doteq \lambda|\tilde{w}\succ .$$

2. Complex eigenvalues $\lambda$ come in complex conjugate pairs, $\lambda_+$, $\lambda_-$ such that $\lambda_- = (\lambda_+)^*$. The corresponding eigenstates,

$$|w_+\succ \doteq \begin{pmatrix} u_+(z) \\ v_+(z) \end{pmatrix}$$
$$|w_-\succ \doteq \begin{pmatrix} u_-(z) \\ v_-(z) \end{pmatrix} ,$$

are related by

$$u_-(z) = -(v_+)^*(z)$$
$$v_-(z) = +(u_+)^*(z) . \tag{19}$$

Here,

$$\hat{\mathcal{L}}|w_+\succ \doteq \lambda_+|w_+\succ$$
$$\hat{\mathcal{L}}|w_-\succ \doteq \lambda_-|w_-\succ$$
$$\lambda_- = (\lambda_+)^* .$$

Within the context of the Inverse Scattering Transform, the wavefunction $\psi(x, t)$ in the parent NLSE, Eq. (10), is assumed to be localized in space, while the eigenstates of the Lax operator $\hat{\mathcal{L}}$ of Eq. (12) are required to be finite at $x = \pm\infty$. In this case, the real eigenvalues of $\hat{\mathcal{L}}$ form a continuum spectrum, while the complex eigenvalues are discrete. Finally, in the parent NLSE, the complex eigenvalues correspond to the solitonic part of the scattering data, while the real eigenvalues correspond to the thermal noise.

From now on, we will assume that the eigenstates of $\hat{\mathcal{L}}$ with substantially complex eigenvalues (i.e. the "discrete spectrum", or "bound states") are normalized as

$$\prec w_-|w_+\succ = 1 . \tag{20}$$

For the case of a single soliton,

$$\psi(z) = -i \operatorname{sech}(x) ,$$

the corresponding eigenvalues and eigenstates are

$$\lambda_+ = -\frac{i}{2}$$
$$|w_+\succ \doteq \frac{+i}{2} \exp(x/2) \begin{pmatrix} -1 + \tanh(x) \\ \operatorname{sech}(x) \end{pmatrix}$$

and

$$\lambda_- = +\frac{i}{2}$$

$$|w_-\succ \doteq \frac{-i}{2}\exp(x/2)\begin{pmatrix} -\operatorname{sech}(x) \\ -1+\tanh(x) \end{pmatrix}.$$

## H.2    The Hellmann-Feynman theorem

Let $\hat{\mathcal{L}}$ depend on a parameter $\xi$: $\hat{\mathcal{L}} = \hat{\mathcal{L}}(\xi)$. Its discrete eigenvalues $\lambda_\pm$ and the corresponding eigenstates, $|w_\pm\succ$ then also depend on $\xi$: $\lambda_\pm = \lambda_\pm(\xi)$, and $|w_\pm\succ = |w_\pm(\xi)\succ$.

Let us express the eigenvalue $\lambda_+$ as

$$\lambda_+(\xi) = \prec w_-(\xi)|\hat{\mathcal{L}}(\xi)|w_+(\xi)\succ .$$

Then, the derivative of $\lambda_+$ with respect to $\xi$ becomes

$$\frac{d}{d\xi}\lambda_+ = \left(\frac{d}{d\xi}|w_-\succ, \hat{\mathcal{L}}|w_+\succ\right) + \left(|w_-\succ, \left(\frac{d}{d\xi}\hat{\mathcal{L}}\right)|w_+\succ\right) + \left(|w_-\succ, \hat{\mathcal{L}}\frac{d}{d\xi}|w_+\succ\right)$$

As in the proof of the usual Hellmann-Feynman theorem, the sum of the first and the last term will turn out to be proportional to the derivative of the norm; and since the norm is one, the derivative is zero. Indeed, the first term gives

$$\left(\frac{d}{d\xi}|w_-\succ, \hat{\mathcal{L}}|w_+\succ\right) = \left(\frac{d}{d\xi}|w_-\succ, \lambda_+|w_+\succ\right)$$

$$= \lambda_+\left(\frac{d}{d\xi}|w_-\succ, |w_+\succ\right).$$

Similarly, the last term gives

$$\left(|w_-\succ, \hat{\mathcal{L}}\frac{d}{d\xi}|w_+\succ\right) = \left(\hat{\mathcal{L}}|w_-\succ, \frac{d}{d\xi}|w_+\succ\right)$$

$$= \left(\lambda_-|w_-\succ, \frac{d}{d\xi}|w_+\succ\right)$$

$$= (\lambda_-)^*\left(|w_-\succ, \frac{d}{d\xi}|w_+\succ\right).$$

But $(\lambda_-)^* = \lambda_+$; thus, the sum of the two terms gives

$$\lambda_+\left[\left(\frac{d}{d\xi}|w_-\succ, |w_+\succ\right) + \left(|w_-\succ, \frac{d}{d\xi}|w_+\succ\right)\right] = \lambda_+\frac{d}{d\xi}\left(|w_-\succ, |w_+\succ\right) = \lambda_+\frac{d}{d\xi}1 = 0.$$

Thus, we get the following generalization of the Hellmann-Feynman theorem:

$$\frac{d}{d\xi}\lambda_+ = \prec w_-|\left(\frac{d}{d\xi}\hat{\mathcal{L}}\right)|w_+\succ . \tag{21}$$

### H.3 The exact expression for $d\lambda/dt$ from the Hellmann-Feynman theorem

Let us set

$$\xi = t\,,$$

$$\hat{\mathcal{L}}(t) = \begin{pmatrix} -i\frac{\partial}{\partial z} & \psi^*(z,\,t) \\ -\psi(z,\,t) & +i\frac{\partial}{\partial z} \end{pmatrix}\,,$$

$$\hat{\mathcal{L}}(t)|w(t)\rangle\!\!\succ = \lambda(t)|w(t)\rangle\!\!\succ\,,$$

$$\frac{d}{dt}\hat{\mathcal{L}}(t) = \begin{pmatrix} 0 & +(F[\psi] + \epsilon P[\psi])^*(z,\,t) \\ -(F[\psi] + \epsilon P[\psi])(z,\,t) & 0 \end{pmatrix}\,,$$

$$|w(t)\rangle\!\!\succ = \begin{pmatrix} u(z,\,t) \\ v(z,\,t) \end{pmatrix}\,,\text{ and}$$

$$2\int_{-\infty}^{+\infty} u(z,\,t)v(z,\,t) = 1\,,$$

where $F[\psi](z,\,t) = -i\left[-\frac{\partial^2}{\partial z^2} - 2|\psi(z,\,t)|^2\right]\psi(z,\,t)$, and $P[\psi](z,\,t) = -iv_{ext}(z,\,t)\psi(z,\,t)$. According to the Hellmann-Feynman theorem, Eq. (21), the time derivative of the Lax eigenvalue is given by

$$\frac{\partial}{\partial t}\lambda = -\int_{-\infty}^{+\infty} dz \left\{u_+^2(z,\,t)\epsilon P[\psi](z,\,t) - v_+^2(z,\,t)\epsilon^* P^*[\psi](z,\,t)\right\}$$

$$= (+i)\int_{-\infty}^{+\infty} dz \left\{\epsilon u^2(z,\,t)\psi(z,\,t) + \epsilon^* v^2(z,\,t)\psi^*(z,\,t)\right\} v_{ext}(z,\,t)\,.$$

Notice that the contribution to $d\lambda/dt$ from $F[\psi]$ disappears. Indeed this contribution describes the time derivative of the Lax eigenvalue in the time evolution according to the *unperturbed* NLS; this derivative indeed vanishes as a consequence of integrability of the NLSE.

A translation to the Kivshar-Malomed notation system of Ref. [27] gives

$$\frac{\partial}{\partial t}\lambda_n = -\frac{1}{2}\frac{1}{\int_{-\infty}^{+\infty} dz\,\psi^{(1)}(x,\,t)\psi^{(2)}(x,\,t)}$$

$$\times \int_{-\infty}^{+\infty} dz \left\{(\psi^{(1)})^2(x,\,t,\,\lambda_n)\epsilon P[\psi](x,\,t) - (\psi^{(2)})^2(x,\,t)\epsilon^* P^*[\psi](x,\,t)\right\}\,.$$

# I  Supplementary Figures

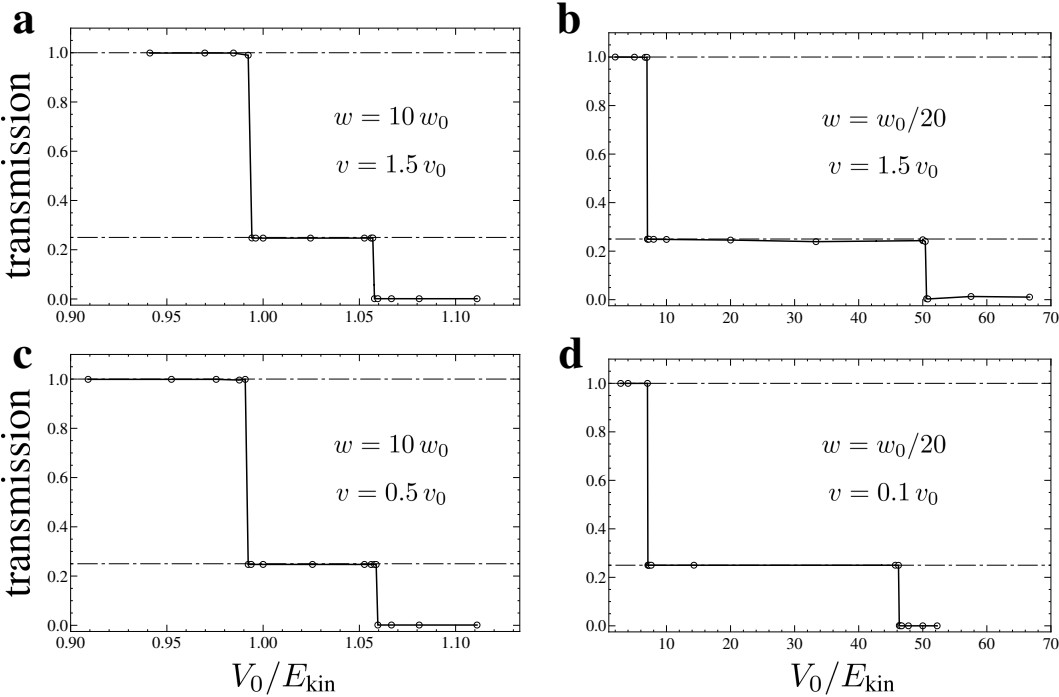

Figure 4: **Preservation of constituent solitons in different regimes.** The same numerical experiment as in Fig. 1 (and the solid line of Fig. 2) of the main text, but with different impact velocities $v$ and barrier widths $w$. Here $v_0 = 0.025$ and $w_0 = 9.5$ (in natural units) are, respectively, the impact velocity and barrier width from Fig. 1 of the main text.

Note that velocities are not the same in **c** and **d**. The reason for this is that when the barrier is wide and the impact velocity slow ($w = 10\,w_0$ and $v = 0.1v_0$), the intermediate step where the transmission coefficient is 1/4 is so narrow that we were unable to resolve it. Qualitatively, in this case we observe that the breather as a whole is either completely reflected or completely transmitted, with no discernible separation of the constituent solitons. The threshold is at about $V_0/E_{\rm kin} = 1.017$. The same happens when $w = w_0$ but $v = 0.01v_0$, with the threshold at $V_0/E_{\rm kin} = 1.626$. This regime of 'complete preservation' of the breather seems qualitatively different enough from the regimes studied thus far that we decided to plot in **c** a situation where the impact velocity is somewhat higher and the intermediate step is clearly visible.

The consoliton preservation is visibly imperfect for $v = 1.5v_0$ (**a** and **b**), because in this case the ratio that governs how well the consolitons are energetically protected against shedding particles during collision, $(E_{\rm rest}/N_{\rm a})$, is only abut 3.7 for the the smaller consoliton. This should be compared to about 8 when $v = v_0$ and to about 33 when $v = 0.5v_0$.

Nevertheless, the effect of consoliton preservation is still clearly operational in all four cases plotted here, and so it holds over a factor of 15 of variation in impact velocity (so a factor of $15^2 = 225$ of variation in the ratio $(E_{\rm rest}/N_{\rm a})/E_{\rm K,1}$) and a factor of 200 of variation in barrier width. Note that when $w = 10\,w_0$, the barrier is an order of magnitude wider than the typical size of the breather,

whereas when $w = w_0/20$ the barrier is a factor of 17 narrower than narrowest feature of the breather (the central peak at half-period in Fig. 3).

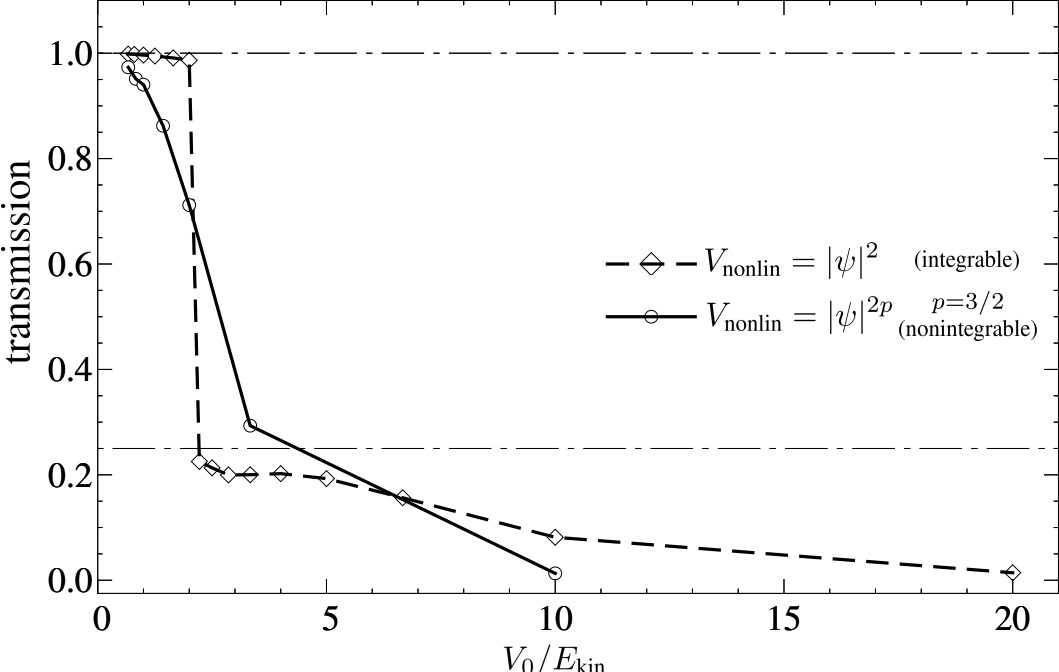

Figure 5: **Integrable vs. nonintegrable case.** The initial state for the nonintegrable case was prepared by time-propagating the breather at rest while the nonlinearity was slowly ramped from $|\psi|^2$ to $|\psi|^{2p}$ with $p = 3/2$. The result was Galilei-boosted and scattered off of a Gaussian barrier whose width (for numerical reasons) was twice the reference value (i.e. twice the experimental value quoted in the main text). All other parameters were at their reference values, including the number of particles. Also plotted is the integrable case, all of whose parameters are at their reference values. In particular, the number of particles is four times smaller than that for Fig. 3 in the main text, resulting in a degraded, but still noticeable plateau at around 25% transmission. In the nonintegrable case, no plateau can be discerned.

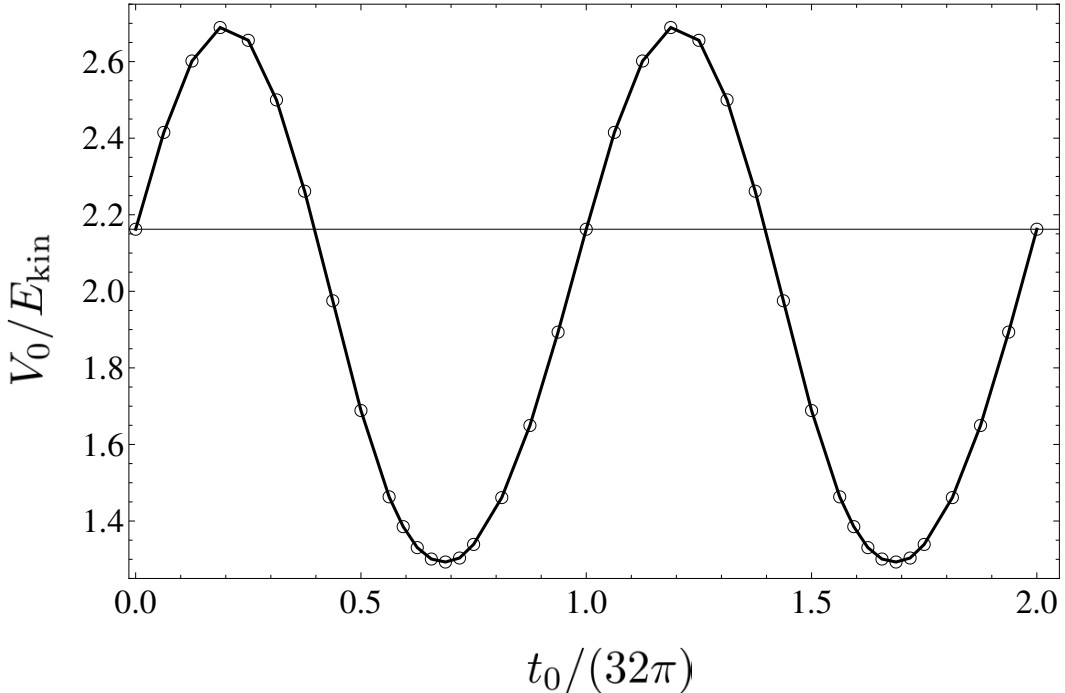

Figure 6: **Dependence on the phase of the breathing cycle.** The $y$-axis: the value of $V_0/E_{\text{kin}}$ for which transmission jumps from 1/4 to 0; the $x$-axis: the time offset in the breathing cycle for the initial state, relative to that in Fig. 2 in the main text; all other parameters are as in that Figure.

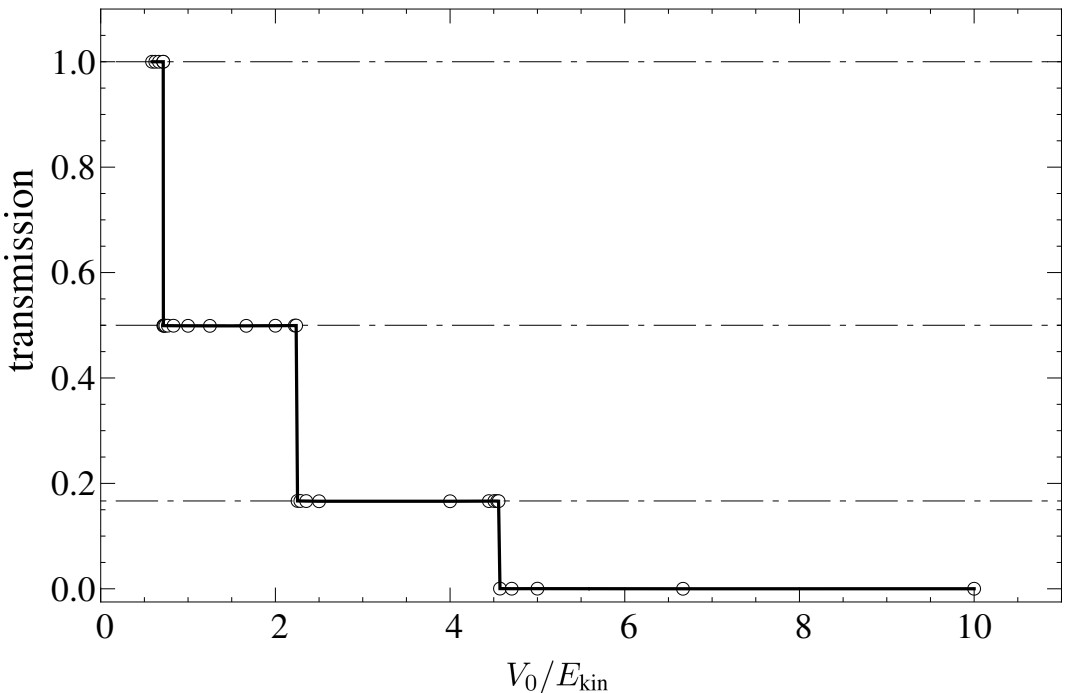

Figure 7: **Transmission plot for a three-soliton breather solution.** The constituent solitons have norms 1/6, 1/3, and 1/2 (1:2:3 norm ratio, which does *not* belong to the sequence of odd number ratios). The dash-dotted horizontal lines are at transmissions values of 1/6, 1/2, and 1, corresponding, respectively, to only the smallest, norm-1/6 soliton being transmitted, to the norm-1/6 and norm-1/3 solitons being transmitted, and to all three constituent solitons being transmitted.

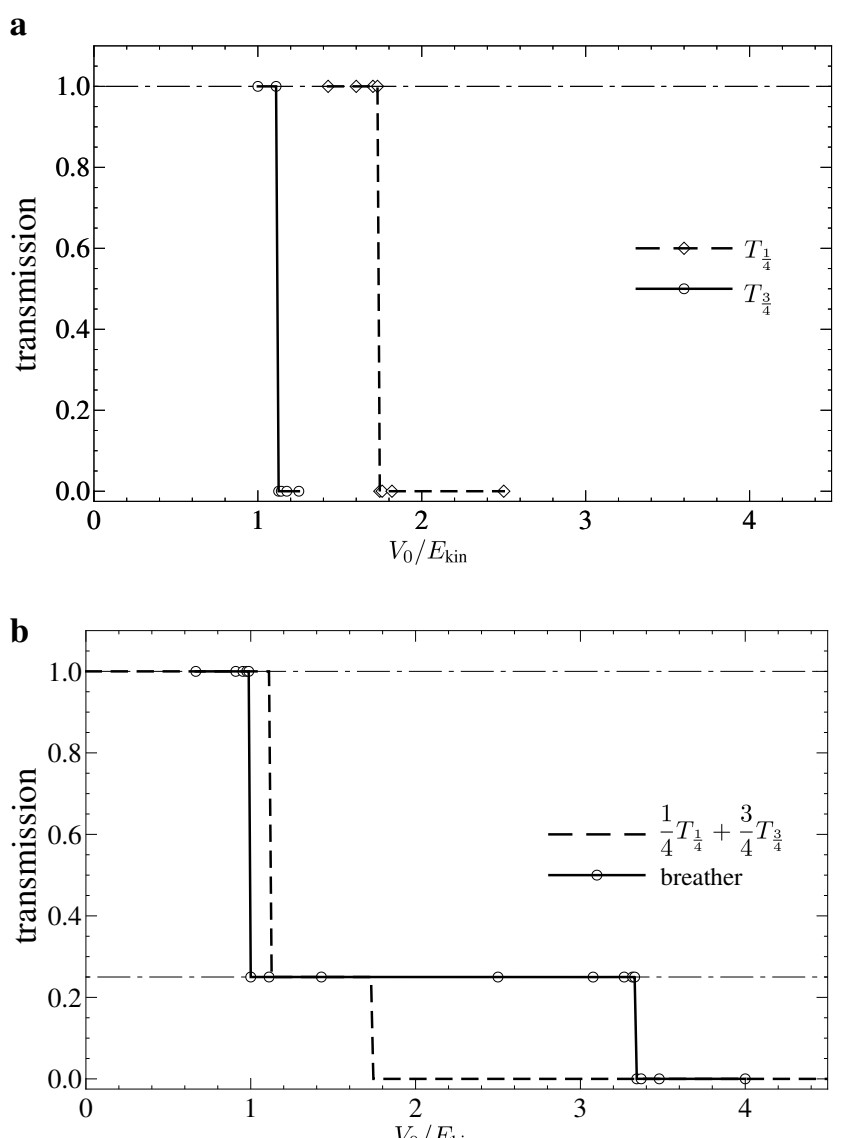

Figure 8: **Uncoupled solitons vs. breather.** How the "staircase" plot of Fig. 2 in the main text would look if the constituent solitons of the breather were completely uncoupled, as compared to now it is in reality. **a,** The transmission plots for the scattering of single solitons, of norms $1/4$ and $3/4$, off a barrier. All parameters are as in Fig. 2 in the main text, with barrier width $w = w_0$. **b,** dashed line: the weighted sum of the single-soliton transmission curves from panel a, with the norms used as weights. Solid line: the transmission curve for the breather for the same set of parameters. This is the same curve as the $w = w_0$ curve in Fig. 2 in the main text.

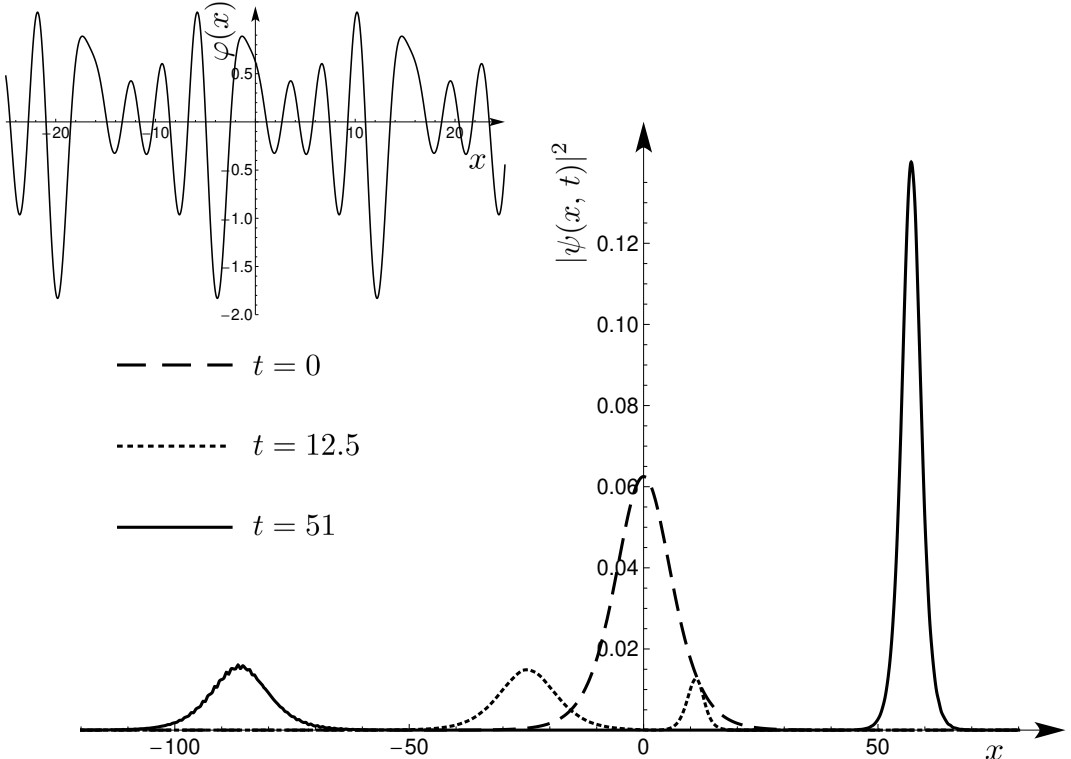

Figure 9: **Superheated integrability in the context of phase imprinting.** At $t = 0$, the breather wavefunction is multiplied by the space-dependent pure phase $e^{i\epsilon\varphi(x)}$. Here $\epsilon = 0.01$ and $\varphi(x) = \sqrt{\frac{2}{L}}\sqrt{\frac{3}{2M}}\sum_{m=1}^{M}[c_m\cos(2\pi mx/L) + s_m\sin(2\pi mx/L)]$, with $L = 16$ and $M = 5$; $c_m$ and $s_m$ were drawn from the uniform distribution on $[-1, 1]$, and in the realization shown here had the values $(c_1, \ldots, c_5) = (0.307, 0.622, 0.648, -0.738, 0.304)$ and $(s_1, \ldots, s_5) = (0.353, -0.0422, -0.794, -0.746, 0.721)$. The function $\varphi(x)$ is plotted in the inset. The main plot shows the time evolution of the density $|\psi|^2$, during which the constituent solitons separate. Once they are well-separated, one can verify that their norms are 0.25 and 0.75. In the context of Eq. (5) in the main text: if one uses the approximation $e^{i\epsilon\varphi(x)} \approx 1 + i\epsilon\varphi(x)$, then the process depicted in this Figure corresponds to $v_{\text{ext.}}(x, t)\psi(x, t) = i\delta(t)\varphi(x)\lim_{\tau \to t^-}\psi(x, \tau)$. Just as in the case of collision with a barrier, the separation of scales between the real and imaginary parts of the Lax-operator eigenvalues $\lambda$, which correspond to the constituent solitons, again imply that the soliton velocities ($\sim \operatorname{Re}\lambda$) change but norms ($\sim \operatorname{Im}\lambda$) do not.

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
