# Peer review of "Resilience of constituent solitons in multisoliton scattering off barriers"

_SciPost Physics_

## Round 3 · Referee Report · Anonymous · 2019-3-4

Strengths

1) An involved non-linear process is dissected in a fairly clear manner.

Weaknesses

1) The presentation misses opportunities provided by 2d graphs or movies.

Report

The article studies the transmission properties of compound solitons, odd-norm-ratio breathers off narrow potential barriers. These are akin to a bound state of a large and a small soliton, with norm ratio 3:1. The authors numerically find that when reducing the barrier height with fixed impact velocity, three regimes are found when (i) both constituent solitons reflect, (ii) both transmit, or (iii) the smaller transmits and the larger reflects. This behavior is then understood also analytically using the inverse scattering transform.

The research presented is interesting and the article fairly well written, so I can recommend publication on SciPost. I would however urge the authors to consider the revisions listed below to improve the presentation and make it somewhat easier to follow.

Requested changes

1) The process considered, soliton scattering off a barrier, can best be understood in a time-dependent picture. It would be helpful if in addition to the snapshots in Fig. 1, the authors were providing 2D (t,x) plots showing the entire dynamics of compound soliton impact on the barrier. Alternatively there could be movies as supplemental material.

2) Throughout the article language is fairly colloquial ("this is a bit different", "so we again have our effect"). This may in places work against precision. Colloquial language should be reduced.

3) I have only one query regarding rigor: Fig. 8 makes the point that one cannot simply understand the transmission behavior of the breather by some weighted average of that of the constituents. However since transmission behavior is strongly dependent on the phase in the breathing cycle as shown in Fig. 6, it is anyway unclear how this could be done via a weighted average? Maybe this discussion can be extended.

---

## Round 3 · Referee Report · Anonymous · 2019-3-11

Strengths

1. Presents an interesting proposal along with reasonable experimental parameters to test the theoretical predictions

2. Suggests a useful application of the result

Weaknesses

1. The text can be difficult to follow with many references to material placed in the appendices

2. There is insufficient justification of the parameters used and whether the same effects could be observed in other similar systems

3. Some typographical errors and inconsistencies in the text

Report

The manuscript explores theoretically what happens when a breather, a nonlinear superposition of multiple solitons, scatters from a repulsive Gaussian potential. The wavepacket is observed to split with the resulting solitons being either reflected or transmitted, dependent of the barrier height, relative to the kinetic energy. Crucially, the norms of the constituent solitons are preserved after the scattering event. The authors propose that this has potentially important applications in soliton interferometry.

I have several questions and comments, outlined below, that I feel need to be addressed before I can recommend the manuscript for publication. This would make the manuscript clearer for the reader.

1. In Fig. 1a the density profile at t=10,500 appears asymmetric. What causes this? Is this due to repulsion from the barrier or the fact the two solitons are moving apart from each other?

2. In Fig. 1 the barrier widths are made 10 times larger than those used in the experiment. In the original experiment the barrier size was comparable to the soliton width. Does this really have no impact as now the barrier width is >> the soliton sizes? Is the system now in a classical regime or where would this limit be? Section 3 suggest that the barrier width is between the substantially quantum and semiclassical regimes but is this still true if the width is >> soliton size?

3. In section 2 the 1:3 breather is considered. Are the authors able to comment on whether the results hold for other ratio breathers? Experimentally there may be fluctuations in the scattering lengths obtained due to uncertainty in the magnetic field. Will this affect the possibility to observe the splitting? Although lithium has a broad Feshbach resonance typically used in these experiments, solitons realised in other alkali atoms, i.e. Rb and Cs, do not use such broad features and so are more sensitive to field fluctuations. Although the atom number per soliton and imaging resolution might put a practical limit on what values of scattering length could be used, it would be useful to know if the splitting effect is a general one.

4. Section 3 refers to ‘our numerics’ but it is not clear what method is used for the numerical calculations. Could some brief details be added here?

5. In section 4 the scaling argument for why the soliton norm is protected is compelling but is given for the case where the breather is wider than the barrier. Why choose this scenario when the numerics presented in the main body of the paper seem to concern the inverse case?

6. In section 5, the experimental proposal is to create the initial breather by jumping the scattering length. Could the authors comment about how this is different to the results presented in Weiss and Carr, arXiv 1612.05545v1 which suggests that, in the quantum many body description, this protocol results in a single soliton surrounded by a thermal cloud.

7. In section 5 it is suggested that the breather might be ‘metastable more robustly’ than a regular soliton. What does this mean?

8. The use of the term ‘superheated’ seems forced and it is not especially clear where the analogy of heating comes from. Could this be elaborated on.

Requested changes

1. In the introduction, soliton collisions with barriers are discussed but only for the repulsive case. There is no reference to any literature concerned with scattering from attractive potentials, either theoretically or experimentally, e.g. Ernst and Brand, Phys. Rev. A 81, 033614 (2010), Marchant et al. Phys. Rev. A 93, 021604(R) (2016). For completeness, these or similar, should be included.

2. In Fig.1, at what time does the breather reach and overlap with the barrier and what is the breathing period in units of ‘t’? This should be added to the figure caption.

3. What is the value of $w_0$ in figure 2? Does it relate to the breather width at all?

4. At the beginning of section 4 the function $u$ and $v$ are introduced. $v$ is later defined to be the consoliton velocity but there is no definition of $u$. Is there any physical meaning to $u$ that could be added?

5. Section 4, pg. 7, 2nd paragraph, the text says that the soliton norm is the imaginary part of lambda and the velocity the real part of lambda. This seems to be reversed compared to the inline equation on page 6., $\lambda _j = A_j/2 + i v_j$. This is also repeated in paragraph 5 on page 7.

6. The sentence at the start of the second paragraph on page 7 ‘Superheated integrability…’ does not make sense. This should be corrected.

7. In appendix A there is a typo/ formatting issue in Eq. 4 on the interaction term.

8. The start of Appendix I is somewhat abrupt. I assume this refers to Fig. 4 of the supplementary figures but this is not explicit.

---

## Editorial Decision

editor-in-charge_assigned